# MD Simulation Reveals Regulation of Mechanical Force and Extracellular Domain 2 on Binding of DNAM-1 to CD155

**DOI:** 10.3390/molecules28062847

**Published:** 2023-03-21

**Authors:** Liping Fang, Yang Zhao, Pei Guo, Ying Fang, Jianhua Wu

**Affiliations:** Institute of Biomechanics, School of Biology and Biological Engineering, South China University of Technology, Guangzhou 510006, China

**Keywords:** DNAM-1, CD155, molecular dynamics simulation, structure-function relation, mechano-chemical signaling

## Abstract

Two extracellular domains of the adhesive receptor DNAM-1 are involved in various cellular biological processes through binding to ligand CD155, usually under a mechano-microenvironment. The first extracellular domain (D1) plays a key role in recognition, but the function of the second extracellular domain (D2) and effects of force on the interaction of DNAM-1 with CD155 remain unclear. We herein studied the interaction of DNAM-1 with CD155 by performing steered molecular dynamics (MD) simulations, and observed the roles of tensile force and D2 on the affinity of DNAM-1 to CD155. The results showed that D2 improved DNAM-1 affinity to CD155; the DNAM-1/CD155 complex had a high mechanical strength and a better mechanical stability for its conformational conservation either at pulling with constant velocity or under constant tensile force (≤100 pN); the catch–slip bond transition governed CD155 dissociation from DNAM-1; and, together with the newly assigned key residues in the binding site, force-induced conformation changes should be responsible for the mechanical regulation of DNAM-1′s affinity to CD155. This work provided a novel insight in understanding the mechanical regulation mechanism and D2 function in the interaction of DNAM-1 with CD155, as well as their molecular basis, relevant transmembrane signaling, and cellular immune responses under a mechano-microenvironment.

## 1. Introduction

Cytotoxic T lymphocyte (CTL) and natural killer (NK) cells play an important role in preventing the occurrence and development of tumors, as well as the infection of viruses and bacteria [1,2]. Recognition of tumor cells, and activation of CTL and NK cells are jointly completed by antigen receptors and various adhesion molecules [3,4,5]. As a key member of the immunoglobulin superfamily, DNAX accessory molecule 1 (DNAM-1), which is widely expressed in most T cells, NK cells, activated platelets, and endothelial cells, is deeply involved in the above cellular immune responses through binding with its ligand CD155, a type I transmembrane glycoprotein in the Ig superfamily [6,7,8,9]. CD155, also called Poliovirus receptor (PVR), is widely expressed on epithelial cells, endothelial cells, nerve cells, and fibroblasts, and is upregulated in some human malignancies such as colon cancer and melanoma [10,11]. DNAM-1 on NK cells interacts with tumor-expressed CD155, promoting NK cells to adhere to tumor cells and induce the cytotoxicity of NK cells [12]. Additionally, the mechanical force also plays a crucial role in the growth and metastasis of tumor cells [13].

DNAM-1 contains two extracellular IgV-like domains, D1 and D2 (Figure 1A) [14]. The first extracellular subdomain (D1) of human DNAM-1 is mainly composed of nine β-sheets, including B, E, D, A, G, F, C, C’, and C” β-sheets, and a short α1-helix between the E and F β-sheets, and all components except the C” β-sheet in D1 are included in the second extracellular subdomain (D2) of the human DNAM-1, while the short α-helix between the E and F β-sheets in D2 is named as the α2-helix (Figure 1A). The two IgV-like domains are stabilized by three disulfide bonds, one in D2′s C-terminal links the C and E β-sheets, and other two connect the B and F β-sheets in D1 and D2, respectively. Additionally, the extracellular domain of CD155 includes three IgV-like domains (D1, D2, and D3), among which the D1 domain interacts with DNAM-1 to trigger subsequent cell–cell (or ECM) cross-talking [14,15]. DNAM-1 interaction with CD155 is based on a conserved lock–key motif within each of their respective D1 domains, resulting in the concentrating of the complex interface on the GFCC’C” β-sheets and some loops of the DNAM-1 and CD155 (Figure 1) [14].

D1 of DNAM-1 is essential for binding with CD155, but the binding efficiency will be significantly reduced in the absence of DNAM-1 D2 [16,17]. It suggests a cooperativity between D1 and D2 of DNAM-1 in binding to CD155 [17]. Furthermore, the mechanical force also plays a crucial role in the growth and metastasis of tumor cells [13]. The mechano-sensitive proteins, such as integrins, selectins, and other transmembrane receptors/ligands, have been reported to have the force-induced changes of conformations and functions [18,19,20,21]; these changes will enhance or reduce subsequent transmembrane signaling, cell–cell interactions, and cellular biological processes [19,22,23]. It hints that a force-dependent interaction between DNAM-1 and CD155 may be required in mediating the involved cellular events in mechano-microenvironment, such as substrate rigidity, blood vessel extension, blood flow shear force, and so on [19,24,25]. However, the role of D2 and the mechanical regulation of DNAM-1′s interaction with CD155 remain unclear.

We herein studied the mechano-regulation and structural basis for the interaction between DNAM-1 and CD155 using steered molecular dynamics (SMD) simulation. The crystal structure of the DNAM-1/CD155 complex (PDB ID: 6O3O) was used to construct two systems (Figure 1A,B) in which the complex was either the wild type (WT), consisting of the DNAM-1 (19th–241st residues) and CD155 D1 (27th–145th residues), or the resulting mutant one, named as ΔD2, from the WT complex with a DNAM-1 D2 deletion. Our results suggested a possible mechanical regulation mechanism and its structural basis of DNAM-1/CD155 interaction might be helpful for deeply understanding the DNAM-1/CD155 interaction which is involved cellular physiological and pathological processes, and provide the useful structural information for novel DNAM-1 or CD155-targeted antibody drug design.

## 2. Results and Discussion

### 2.1. The Second Extracellular Domain D2 Improved DNAM-1 Affinity to CD155

To investigate the role of DNAM-1 D2 on the interaction between DNAM-1 and CD155, we first examined DNAM-1 affinity to CD155 in the presence and absence of DNAM-1 D2 via thrice 60 ns system equilibrium simulations using two different molecular systems, in which the wild type (WT) and the mutant complexes were used to build the complex models, respectively. The WT complex was taken from the crystal structure of the DNAM-1/CD155 complex (PDB ID: 6O3O) consisting of the DNAM-1 (19th-241st residues) and CD155 D1 (27th-145th residues), and the mutant one, named as ΔD2, was the DNAM-1 D2-deleted WT complex (Figure 1A,B).

The results showed that the mean C-RMSD of the WT complex was slightly larger than that of the resulting mutant complex ΔD2 (Figure 2A and Appendix A), meaning that the presence of DNAM-1 D2 might reduce the Brownian-motion-derived variation of the complex structure and make the complex stable. The dissociation probability $f_{D}$ of the WT complex was significantly lower than that of the ΔD2 (Figure 2B), possibly coming from the stronger hydrogen bonding across a larger interfacial surface (Figure 2C,D) and a smaller binding energy (*E*) (Figure 2E) of the WT complex rather than the ΔD2 complex. In fact, the numbers of hydrogen bonds (N_HB_) on the interfaces of the WT and ΔD2 complexes were read to be 8.9 ± 2.4 and 6.8 ± 1.7 (mean ± S.D), respectively (Figure 2C and Appendix A), and the mean buried SASA of the WT and ΔD2 complexes were 1680.2 ± 174.0 and 1508.5 ± 145.3 Å2, respectively (Figure 2D), while the mean binding energies (E) of the WT and ΔD2 complexes were −232.6 ± 92.5 and −144.3 ± 64.6 kcal/mol, respectively (Figure 2E). These data indicated that the presence of D2 upregulated DNAM-1 affinity to CD155 through enhancing the interfacial hydrogen bonding, enlarging the binding surface, reducing the binding energy, and promoting the structural stability of the WT complex. This result was consistent with the experimental data that the efficiency of DNAM-1 D1 binding to CD155 was significantly reduced after the removal of D2 [17].

Then, we detected twelve H-bonds on the binding site through 60 ns system equilibriums thrice, and found that D2 of DNAM-1 enhanced all the twelve interfacial H-bonding events except the bonding of Q47 on DNAM-1 to S62 on CD155 (Figure 3 and Table 1). Including the interfacial H-bonds with occupancies (>0.40) in the top seven, the six residues (T46, Q47, E49, T112, Y113, Q115) on DNAM-1 D1, together with their respective partners (S62, Q63, S74, G83, T127, S132) on CD155, should be crucial in the binding of CD155 to DNAM-1 (Figure 3 and Table 1). Of these key residues, Y113 and Q63 were identified through mutation experiments [14,17], and only S74 and its partner Q115 were located at the minor binding subsite in the loop (Figure 3). Paired with E57 on CD155, R72, another possible key residue on DNAM-1, contributed an H-bond with mean occupancies larger than 25%, and might be responsible for the two complexes; R72 and its other partner D28 formed an H-bond with a mean occupancy of 28% for the WT complex but only 4% for the ΔD2 complex; the mean occupancy of the H-bond between R133 and E49 or S59 was nearly 25% for the WT complex but about 8–14% for the ΔD2 complex; and the H-bond between K190 and E71 in WT had a mean occupancy of 23% but was lost in ΔD2 because K190 did not exist in ΔD2 (Table 1 and Figure 3C). These data indicate that the absence of the D2 domain weakens H-bonding across the complex interface.

### 2.2. Dissociation of the Stretched DNAM-1/CD155 Complex Was Biphasic Force-Dependent

We examined the regulation of tensile force on the dissociation of the WT DNAM-1/CD155 complex through performing the “ramp-clamp” steered molecular dynamics (SMD) simulations thrice with a constant pull velocity of 3 Å/ns in the “force-ramp” but under various constant tensile conditions (0, 25, 50, 75 and 100 pN) in the “force-clamp” mode for 100 ns. From the three “force-ramp” runs, we found that the secondary structures of the WT complex were better preserved under pulling with a velocity of 3 Å/ns, in spite of a slight pull-induced change of the structure (Figure 4A). The force–time curves showed that the rupture force of the complex was 200–250 pN, showing a higher mechanical strength or a better ability to resist pull-induced dissociation of the complex (Figure 4B). The plateau of the buried-SASA–time plots showed the mechanical stability of the complex in front of pull-induced dissociation, while the slight fluctuations of the RMSD around its mean level reflected the well-conserved secondary structures of DNAM-1 and CD155, no matter if the complex was dissociated.

In addition, we found the effects of tensile force on DNAM-1 binding to CD155 through the thrice “ramp-cramp” SMD simulations of 100 ns under various tensile forces from 0 to 100 pN. The time courses of the Cα-atom of RMSD and the distance from the pull atom to the fixed one (Figure 5A,B) showed that the Cα-atom RMSD varied within a range below 4.5 Å, exhibiting a stable conformation of the stretched complex, and the distances between the steered and fixed atom stayed at a plateau with a slight roughness of a height of 5 Å, suggesting a fully allosteric modulation of the stretched complex. The Gaussian fitting of the interfacial H-bond number frequency meant that the present results might be receivable because the sampled space of the data was quasi-complete (Figure 5C). Plots of the mean binding energy (*E*) and the mean interfacial H-bond number (N_HB_) over 100 ns against tensile force showed that *E* decreased first and then increased with tensile force, and the force threshold occurred at 50 pN (Figure 5D), showing a biphasic force-dependent binding affinity of DNAM-1 to CD155; and in contrast, N_HB_ slightly increased first and then decreased with the tensile force, and the force threshold occurred at 25–50 pN (Figure 5E). As a result, increasing tensile force made the normalized complex dissociation probability ($f_{D}$) decrease first until force reached at a threshold of 25 pN, and then increase (Figure 5F), suggesting a catch–slip bond transition in the dissociation of DNAM-1/CD155. This phenomenon of the catch–slip bond transition had been observed through an atomic force microscope (AFM) and biomembrane force probe (BFP) as well as flow chamber experiments for various adhesion molecular systems, such as P-, E-, or L-selectin with PSGL-1 [21,26,27], von Willebrand factor (VWF) with GPIbα [20], and ADMAMTS13 (A Disintegrin and Metalloprotease with thrombospondin motifs-13) [28]. The difference in the force thresholds of *E*, N_HB_, and $f_{D}$ might come from the high sensitivity of $f_{D}$ to the different H-bond occupancies.

However, there might be a significant gap between the results from the MD simulation and the data measured with single molecular tools, such as atomic force microscopy (AFM), optical and magnetic tweezers [29], coming from effects of timescale on predicting ligand–receptor interactions with a timescale of about 0.01–1.00 s by MD simulation of about 100 ns. Regardless of the timescale effect on complex dissociation $f_{D}$, it was expected here that $f_{D}$, the mechano-regulation factor or the normalized complex dissociation probability, should be comparable with experimental data if the conformations sampled from the simulation are perfect [22,24].

### 2.3. Force-Induced Allostery of the DNAM-1/CD155 Complex

The force-induced allostery might be responsible for the catch–slip bond transition of the interaction of DNAM-1 with CD155. To quantify the tension-induced complex allostery, the distance of mass center (DMC) was introduced herein and defined as the distance from the mass center of the loop linker of the C” and D β-sheets in DNAM-1 D1 to the mass center of the loop linker of the B and C β-sheets in CD155 (Figure 6A,B). The mean DMC plots against tensile force showed that increasing force made DMC extended first and then shortened, and the force threshold was located at 50 pN (Figure 6C), similar to the biphasic force regulation on the complex dissociation probability (Figure 5F). While force-induced allostery promoted DNAM-1 affinity to CD155, DMC shortened and DNAM-1 and CD155 approached each other, all in concert (Figure 5F and Figure 6C). It suggested that the force-induced change of DMC might serve as a molecular basis for the catch–slip transition mechanism of the complex dissociation.

Furthermore, we measured κ, the mean angle between the A β-sheet of DNAM-1 D1 and the C’ β-sheet of DNAM-1 D2 over a simulation time of 100 ns thrice under each constant pull force, to further scale the force-induced allostery of the complex (Figure 6A,D). The angle κ increased remarkably first and then decreased with tensile force, and the turning point occurred at the threshold of 75 pN (Figure 6E). This suggested that the DNAM-1 might also undergo polymorphic conformational changes under the force induction. These results meant that the force-induced allostery of DNAM-1 might be responsible for the biphasic force-dependent dissociation of DNAM-1 and CD155 too. This force-induced change of protein structure–function was also obtained in other mechano-sensitive proteins, such as integrins, selectins, VWF, and so on [19,21,26,28].

### 2.4. The Key Residues in the Binding of DNAM-1 to CD155 under Tensile Force

To reveal the molecular basis of force regulation on the interaction of DNAM-1 with CD155, we further examined the interfacial H-bonds and their involved residue pairs of the WT DNAM-1/CD155 complex, and evaluated the occupancies of these H-bonds through thrice 100 ns clamp SMD simulations under tensile forces of 0, 25, 50, 75, or 100 pN (Materials and Methods). Fourteen H-bonds across the DNAM-1/CD155 complex interface and twelve H-bonds across the D1/D2 interface were detected from three runs, and were contributed by the residue pairs on the DNAM-1/CD155 and DNAM-1 D1/D2 interfaces, respectively (Figure 7A and Appendix A).

The results showed that these H-bonding events were multifariously force-dependent and might become strong or weak in response to tensile force, accompanied with the formation of new bonds or the vanishing of old interactions; an H-bonding event under increasing tensile force might remain stable, be force-enhanced (catch bond) or force-weakened (slip bond), or undergo a transition from catch- to slip-bond or slip- to catch (Figure 7). As a result, five residue interaction modes, including the steady, catch–slip, slip–catch, catch–slip–catch, and the slip–catch–slip type, were shown in their respective typical residue interactions (Figure 7 and Appendix A). These H-bond interactions in a force-dependent manner might mediate the force-induced change of conformation and function of DNAM-1 bound with CD155.

The improved ability of the complex to resist mechanical damage might be attributed mainly to the five complex interfacial H-bonds, which had the top five occupancies (>60% about) under various tensile forces for all fourteen H-bonds and were formed between T46 and Q63, Q47 and either S62 or T127, E49 and S132, and Q115-S74; and, R72 and T112 with their respective partners E57 and Q63 formed two H-bonds with moderate but biphasic force-dependent occupancies, suggesting these two residue pairs might be responsible for the catch–slip bond transition mechanism of the complex dissociation (Figure 7A and Appendix A). All the residues mentioned above should be dominant in the binding of DNAM-1 to CD155, no matter if there was a tensile force on the complex. Additionally, of all the twelve detected H-bonds across the D1/D2 interface, the five dominant ones with higher occupancies (>50%) under various tensile forces were contributed by residue pairs, such as H24–L175, T25–D174, S26–I173, E36–R171, and D128–R221 (Figure 7B and Appendix A). The H-bonding between D128 on D1 and R221 on D2 of DNAM-1 was a strong force-induced event, prompting the stability of the DNAM-1 D1–D2 dimer under a tensile force.

## 3. Conclusions

DNAM-1, as a member of the Ig superfamily, participates in tumor recognition and activation by recognizing its ligand CD155, is relevant to many adverse autoimmune diseases, and may serve as a target for the treatment of diseases [30,31,32]. However, less knowledge of the mechanical regulation of the DNAM-1 interaction with CD155 exists, despite the understanding that DNAM-1-related events usually occur at diverse mechanical micro-environments. Furthermore, a well-known function of the first extracellular subdomain D1 of DNAM-1 is believed to be recognizing CD155, but the role of the second extracellular subdomain D2 of DNAM-1 in DNAM-1 D1 binding to CD155 remains unclear [14,16,17].

We herein investigated mechanical regulation and its molecular basis on DNAM-1′s interaction with CD155 by performing a series of MD simulations. The results demonstrated that DNAM-1 D2 would promote the DNAM-1 affinity with CD155 through making the complex conformation compact. The complex had not only a higher mechanical strength to resist force-induced DNAM-1 dissociation from CD155 because of the complex rupture >200 pN but also a better mechanical stability for its structure conservation under pulling with constant velocity or at a constant tensile force. A catch–slip bond transition was assigned to the dissociation of the DNAM-1/CD155 complex under tensile force, being relevant to the force-induced allostery of the complex. Being responsible for the force-regulated interaction of DNAM-1 with CD155, the key residues at the binding site were predicted to be T46, Q47, E49, R72, T112, Y113, and Q115 on DNAM-1 with their respective partners on CD155. The present results provided a novel insight into not only the mechanical regulation mechanism and its molecular basis in the interaction of DNAM-1 with CD155, but also the immune diagnosis and treatment of tumor or autoimmune diseases related to the DNAM-1/CD155 interaction, and might also be helpful for the development of anti-cancer target drugs.

## 4. Materials and Methods

### 4.1. System Setup

The crystal structure of the DNAM-1/CD155 complex was from the Protein Data Bank (PDB ID: 6O3O) [14]. Two initial models, including (i) the WT complex of DNAM-1 D1 and D2 (residues: 19th–241st) with CD155 D1 (residues: 27th–145th) and (ii) the ΔD2 complex of DNAM-1 D1 (residues: 19th–128th) with CD155 D1 (residues: 27th–145th) (Figure 1), were constructed for MD simulations. The missed residues on DNAM-1 D1 (residues: 84th–88th) were added using Schrödinger software, and the missed hydrogen atoms were added by AUTOPSF, a plug-in of visual merchandise design (VMD) [33,34]. Each of the initial structures were soaked with TIP3P water molecules in a rectangular box with walls at least 15 Å away from any protein atom. The Na^+^ and Cl^−^ counter ions at a physiological concentration of 150 mM were added into the water boxes to achieve charge neutrality and to mimic the physiological environment.

### 4.2. MD Simulation

We herein used the NAMD 2.13 program for MD simulations [35]. To investigate the effect of D2 of DNAM-1 on the binding of DNAM-1 to CD155, the system equilibriums were performed on the WT and ΔD2 systems for 60 ns thrice, respectively (Appendix A). The CHARMM27 [36,37] all-atom force field, along with cMAP correction for backbone, particle mesh Ewald (PME) [38] algorithm for electrostatic interaction, and a 12 Å cut off for electrostatic and van der Waals interactions, were used here to perform MD simulations with a periodical boundary condition and time step of 2 fs. The system was minimized firstly at 15,000 steps with heavy or nonhydrogen protein atoms being fixed, and then minimized at 15,000 steps with all atoms free. The energy-minimized systems were heated gradually from 0 to 310 K in 0.1 ns first and then equilibrated thrice for 60 ns with pressure and temperature control. The temperature was held at 310 K using Langevin dynamics, and the pressure was held at 1 atmosphere by the Langevin piston method. The equilibrated structure of the WT complex of DNAM-1/CD155 was then used as the initial conformation for the subsequent steered molecular dynamics (SMD) simulations.

The so-called “ramp-clamp” SMD simulations, a force-clamp MD simulation followed a force-ramp one [18], were performed on the equilibrated system to examine the force-induced conformational changes and force-regulated dissociation of the complex. The C-terminal Cα atom (A241 on D2) of WT DNAM-1 was fixed, and the C-terminal Cα atom of CD155 (P145) was steered along a pulling direction from the fixed atom to the pull atom. The virtual spring, connecting the dummy atom and the steered atom, had a spring constant of 13.90 pN/nm. The complex was pulled over 20 ns (WT) thrice with a time step of 2 fs and a constant velocity of 3 Å/ns, at which the pulling would be contributed to hydrogen bond (H-bond) rupture with conservation of secondary structures of the complex.

Force-clamp simulation was also carried out by taking the WT complex. Once tensile force reached a given value, such as 0, 25, 50, 75, or 100 pN, the SMD simulation was transformed from the force-ramp mode to a force-clamp one, at which the WT complex was stretched with the given constant tensile force for the following 100 ns thrice. Each event of hydrogen bonding under stretching was recorded to examine the involved residues and their functions.

All these selected values of the virtual spring constant, pull velocity, and tensile forces were moderate and could cause mechano-destabilization of the complex either in the early loading phase in “force-ramp” mode or in the entire simulation duration in “force-clamp” mode.

### 4.3. Data Analysis

All analyses were performed with VMD tools [34]. We measured the Cα root mean square deviation (RMSD) and the solvent accessible surface area (SASA) (with a 1.4 Å probe radius) to characterize the conformational change and the hydrophobic core exposure, respectively. A hydrogen bonding event happened once the donor–acceptor distance and the donor-hydrogen–acceptor angle were less than 3.5 Å and 30°, respectively. A salt bridge was defined if the distance between any of the oxygen atoms of acidic residues (Asp or Glu) and the nitrogen atoms of basic residues (Lys or Arg) were within 4 Å. An occupancy (or survival ratio) of a hydrogen bond (H-bond) or a salt bridge was evaluated by the percentage of bond survival time in simulation period. All visual inspections and molecular images were completed by using VMD 1.9.2. Furthermore, *f_D_*, the probability of ligand dissociation from its receptor, was used herein to approximately evaluate affinity of the receptor to its ligand, as in our previous works [34]. The dissociation was calculated by survival ratios of hydrogen bonds across the binding site, with the assumption that each hydrogen bonding event on the binding site was independent and fully responsible for the DNAM-1 affinity to CD155.

## Figures and Tables

**Figure 1 molecules-28-02847-f001:**
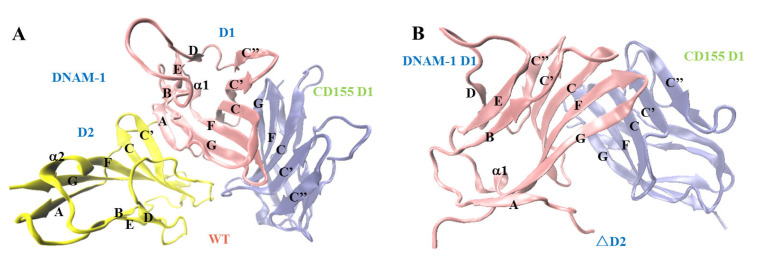
The complex of DNAM-1 and CD155. (**A**) The crystal structure of the wild type (WT) complex of DNAM-1 and CD155 (PDB ID: 6O3O) in new cartoon mode. In the WT complex, the bound DNAM-1 contains the first and second extracellular IgV-like subdomains, D1 (red) and D2 (yellow), but the ligated CD155 just contains the first extracellular IgV-like subdomain D1 (blue). (**B**) The structural model of the resulting mutant complex (ΔD2), which consists of CD155 D1 (blue) and DNAM-1 D1 (red) and is built up by deleting the DNAM-1 D2 from the WT complex.

**Figure 2 molecules-28-02847-f002:**
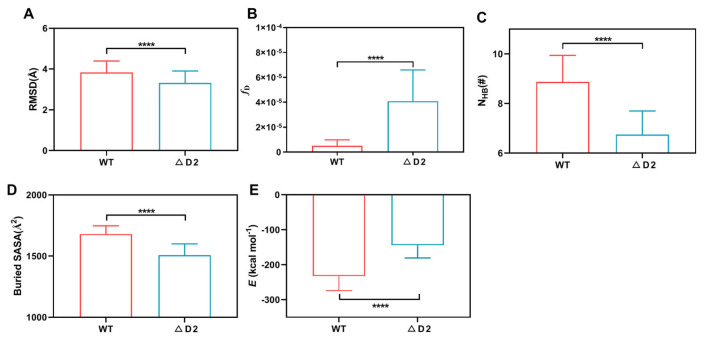
Effect of DNAM-1 D2 on DNAM-1 affinity to CD155. (**A**) The mean C_α_-RMSD, (**B**) the dissociation probability ($f_{D}$), (**C**) the mean interface H-bond number (N_HB_), (**D**) the mean buried SASA, and (**E**) binding energy (*E*) for the WT and ΔD2 complexes over 60 ns equilibrium. The *p*-values of the unpaired two-tailed Student’s *t* test were shown to indicate the statistical difference significance (**** *p* < 0.0001), or lack thereof. Data were shown with mean ± S.D of three runs.

**Figure 3 molecules-28-02847-f003:**
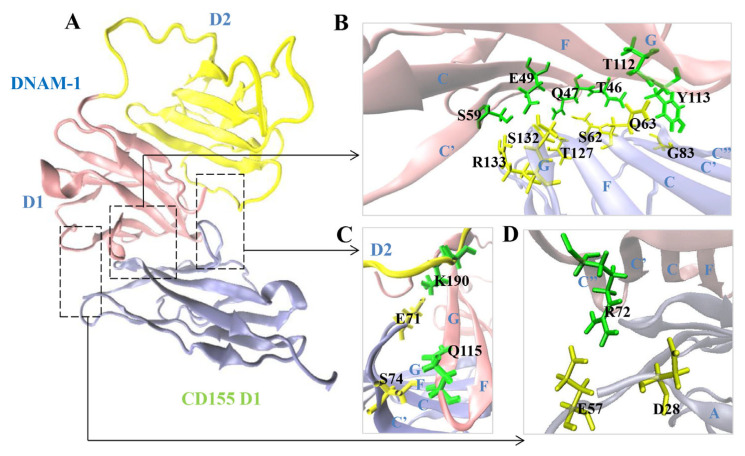
Binding site and its three subsites, the crucial and two minor ones, for the WT complex of CD155 with CD155. (**A**) Overview of the WT complex structure. DNAM-1 (D1, red; D2, yellow) bound to CD155 (blue) was shown in a new cartoon mode. (**B**) The crucial binding subsite was in sheets, and (**C**,**D**) the two minor binding subsites were in the loop. The involved binding residues were labeled in green (DNAM-1) or yellow (CD155) licorice. K190 was located in DNAM-1 D2, paired with E71, and did not exist in the resulting mutant complex ΔD2.

**Figure 4 molecules-28-02847-f004:**
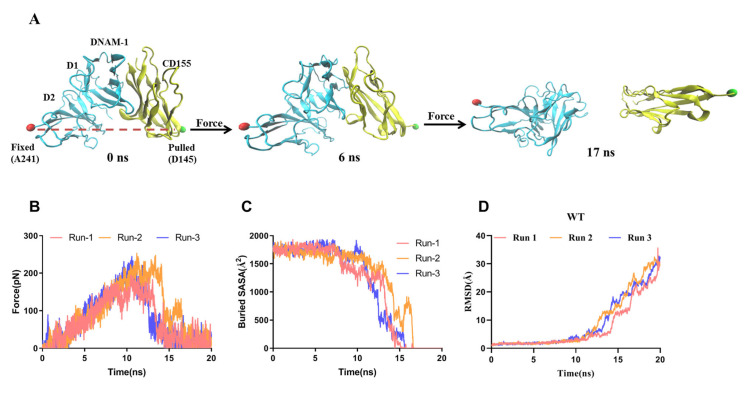
Pull-induced dissociation of the WT complex. (**A**) Three typical snapshots of the WT complex at the stimulus times of 0, 6, and 17 ns in a “force-ramp” SMD simulation with pulling velocity of 3 Å/ns. The C-terminal Cα-atom (A241, red) of DNAM-1 D2 was fixed, while the C-terminal Cα-atom (P145, green) of CD155 was steered along a pulling direction from fixed atom to the pulled one. The time courses of Force (**B**), the buried SASA (**C**,**D**) the RMSD of the WT complex under pulling with velocity of 3 Å/ns from thrice dependent runs.

**Figure 5 molecules-28-02847-f005:**
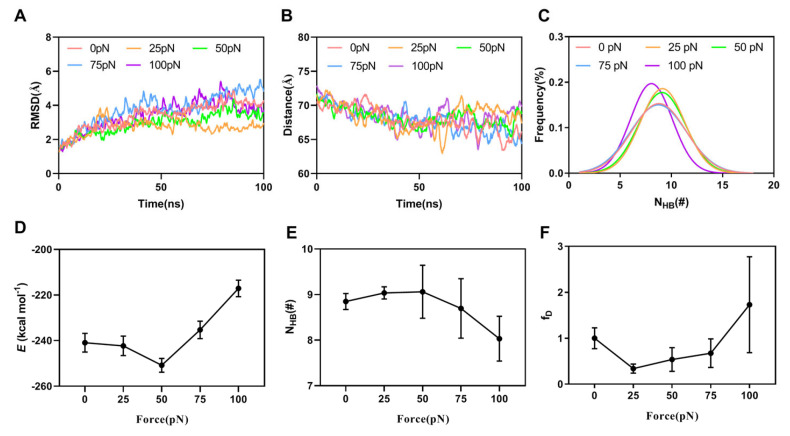
The force-dependent binding affinity of DNAM-1 to CD155. The representative time courses of (**A**) the Cα-RMSD and (**B**) the distance between the steered atom and the fixed one for the WT complex under stretching with 0, 25, 50, 75, and 100 pN, and with time. (**C**) The frequency of the interfacial H-bond number for 100 ns simulations under various tensile forces. Each curve is the mean of the three runs under the corresponding force. Plots of (**D**) the binding energy (*E*), (**E**) the interfacial H-bond number (N_HB_), and (**F**) the dissociation probability $f_{D}$ against tensile force. All data were mean ± S.E. of three runs.

**Figure 6 molecules-28-02847-f006:**
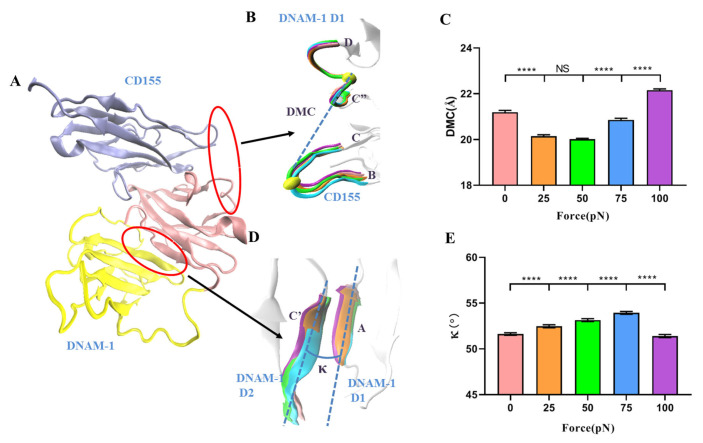
Force-induced change of DMC and the cross-angle between βA in D1 and βD in D2 of DNAM-1. (**A**) Schematic diagram of allosteric DNAM-1/CD155 complex. (**B**) Superposition of five representative conformations of the loop (residues: 72nd–79th) between C” and D β-sheets in DNAM-1 D1 and the loop (residues: 53rd–59th) between the B and C β-sheets in CD155 under tensile forces of 0 (pink), 25 (orange), 50 (green), 75 (blue), and 100 pN (purple). The two yellow spheres expressed the locations of the two loop mass centers. The CD155 were aligned well and are displayed in a new cartoon representation. (**C**) Plots of DMC against tensile force. (**D**) Superposition of five representative conformations of the A β-sheet of DNAM-1 D1 and the C’ β-sheet of DNAM-1 D2 under tensile force of 0 (pink), 25 (orange), 50 (green), 75 (blue), and 100 pN (purple), where the A β-sheet aligned very well and is shown in a new cartoon representation. The angle is quantified by the cross angle of the A and C’ β-sheets. (**E**) Plots of angle against tensile force. The *p*-values of Tukey’s multiple comparisons test are shown to indicate the statistical difference significance (**** *p* < 0.0001), or lack thereof. All data shown are means ± S.E., n = 3.

**Figure 7 molecules-28-02847-f007:**
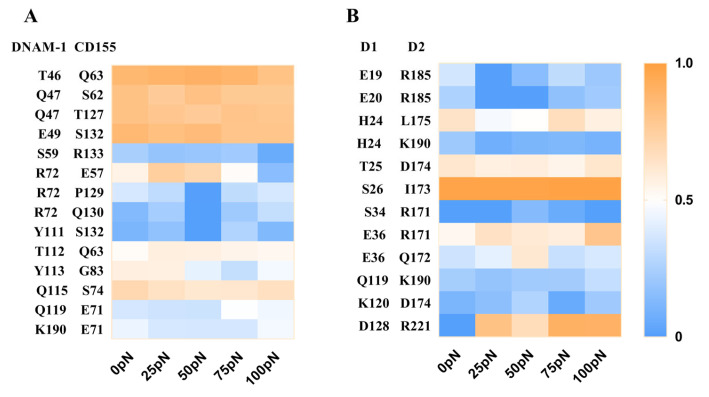
Thermograph for residue interactions on the interfaces of DNAM-1/CD155 and DNAM-1 D1/D2 under various tensile forces. (**A**) The occupancies of fourteen interfacial H-bonds between the respective residue pairs on the complex site binding. (**B**) The twelve H-bonds on the interfacial surface of DNAM-1 D1 and D2, their occupancies, and the involved residue pairs. Each data from three force-clamp SMD runs of 100 ns under each given tensile force. The color bar marked the values of the mean H-bond occupancies. The H-bond occupancy patterns showed the various force-dependent residue interactions.

**Table 1 molecules-28-02847-t001:** Key H-bonds and their survival rate on the interface of the equilibrated DNAM-1/CD155 complex.

No.	DNAM-1	CD155	WT	ΔD2
	Residue	Residue	Occupancy	Occupancy
1	E49	S132	0.90 ± 0.02	0.86 ± 0.07
2	T46	Q63	0.86 ± 0.07	0.80 ± 0.14
3	Q47	T127	0.78 ± 0.02	0.76 ± 0.05
4	Q115	S74	0.67 ± 0.02	0.45 ± 0.24
5	Q47	S62	0.62 ± 0.33	0.70 ± 0.14
6	T112	Q63	0.49 ± 0.42	0.40 ± 0.19
7	Y113	G83	0.49 ± 0.07	0.44 ± 0.23
8	R72	E57	0.33 ± 0.57	0.26 ± 0.44
9	R72	D28	0.28 ± 0.49	0.04 ± 0.07
10	S59	R133	0.25 ± 0.38	0.08 ± 0.08
11	E49	R133	0.24 ± 0.42	0.14 ± 0.25
12	K190	E71	0.23 ± 0.22	0.00 ± 0.00

## Data Availability

The data is available on request from the author.

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
