# Peer review of "MD Simulation Reveals Regulation of Mechanical Force and Extracellular Domain 2 on Binding of DNAM-1 to CD155"

_molecules, 2023, doi:10.3390/molecules28062847_

Round 1

Reviewer 1 Report

It is a well written manuscript in which the author studied the DNAM-1 (WT,D2)..CD155 interaction using SMD and MD simulation technique. The method looks like is reasonable.

My concern:

1. Are you sure that the 3A/ns is enough slow to reach a proper free energy force

2 In Table 1 several survival rate for WT and D2 has very wide distribution. In avarege we could tell that this is larger  (for larger survival propability) mainly everywhere for D2, except on Q47..S62, T112..Q63. If this survival propability smaller than 0.35 than  it is not exactly true. Can you give the author some explanation for this behaviour?

Author Response

Comment 1. Are you sure that the 3Å/ns is enough slow to reach a proper free energy force?

Answer: Yes, we believed that this pulling velocity is enough slow to reach a proper free energy force, despite that we had not do a direct evaluation. In this study, the selection of the pulling velocity of 3Å/ns was based on that, under pulling with this velocity, the complex conformation remained intact almost for most of stimulation time before rupturing of complex.

Comment 2 In Table 1 several survival rate for WT and D2 has very wide distribution. In average we could tell that this is larger (for larger survival probability) mainly everywhere for D2, except on Q47…S62, T112…Q63. If this survival probability smaller than 0.35 than it is not exactly true. Can you give the author some explanation for this behavior?

Answer: Yes, we agree that, the data for the survival probabilities (<0.35) of H-bonds may not be exactly true. The reason should come not only from the effects of sensitive initial-conformation dependence and stochastic thermodynamic environment on MD simulation but also from the size and completeness of the sampled conformation space. However, for simplicity, in this study, we had considered complex conformation completeness by assuming that the complex was quasi-complete if the H-bond pattern (also see Figure 5C) obeyed Gauss distribution. We believed that, If the complex was quasi-complete, the crucial information for the key residues (involved in H-bonds with occupancies >0.35) predicted may be true.

Reviewer 2 Report

Review Report 

Summary

The paper aims to understand the role of the second extracellular domain (D2) of the adhesive receptor DNAM-1 in its interaction with the ligand CD155 under mechanical stress. The authors used steered molecular dynamics simulations to study the effects of tensile force on the affinity between DNAM-1 and CD155, and observed that D2 improves DNAM-1's affinity to CD155. The paper's main contributions include identifying critical residues in the binding site and proposing a force-induced conformational change as the mechanism for mechanical regulation on DNAM-1 affinity to CD155. The paper's strengths lie in its use of molecular dynamics simulations to provide novel insights into the molecular basis of transmembrane signaling and cellular immune responses under mechano-microenvironment.

General comments

In general, the paper provides a detailed and comprehensive study. The authors used advanced computational techniques to investigate the mechanical regulation mechanism and D2 function on the interaction of DNAM-1 with CD155, which is an important area of research in immunology. The paper is well-organized, with clear headings and subheadings that make it easy to follow. The results are presented in a logical and concise manner, with appropriate figures and tables to support the findings. Overall, the paper is a valuable contribution to the field of immunology research.

A couple of questions I have about the methodology used in the study include:

  1. Can the authors provide more supporting reasons /references for the parameters selected and comment on whether the results/conclusions will differ significantly if a different set of simulation configurations are used? 

  2. Are there experimental validations or simulational validations using a simpler but well-studied system?

Special comments

  1. On page 10, the subsubsection numbers do not match the subsection number. For example, 3.1, 3.2 are under subsection 4. Materials and Methods

  2. There are some grammar errors in the manuscripts. For example, this sentence: “However, less knowledge on (of)mechanical regulation on DNAM-1 interaction with CD155 exists, despite the DNAM-1-related events usually occurs (occur)at diverse mechanical micro-environments”. And there are a few places where present and past tense are used inconsistently. I would suggest the author proofread the manuscript more carefully and correct those errors. 

Author Response

Comment 1. Can the authors provide more supporting reasons /references for the parameters selected and comment on whether the results/conclusions will differ significantly if a different set of simulation configurations are used?

Answer: Great suggestion. In this study, the parameters, such as virtual spring constant and pull velocity as well as tensile forces, were selected based on the considerations as follows:

1, The selected virtual spring constant should made moving of the steered-atom move slow enough for preventing pull-induced mechano-destabilization of the pulled complex especially in the early loading phase.

2, Under pulling with the selected constant velocity, the complex conformation should remain intact almost for most of stimulation time before rupturing of complex.

3, In the “force-clamp” mode, the moderate tensile force could not cause mechano- destabilization of the stretched complex during entire simulation duration, based on the mechano-microenvironment of the complex.

The requirement for complex mechano- stabilization could made one gain a quasi-complete complex sample space. It was only from the quasi-complete sample space of the pulled/stretched complex with the stable conformation, we could detect the gain structure-function relation information.

If the selected parameters, such as pull velocity and tensile force, were overlarge, the pull- or force-induced structure damage would occur at the early simulation stage. As a result, the observed force-induced allostery of complex did not exist in physiological environment, and one could not get a quasi-complete complex conformation sample space, leading to an unrepeatable and distorted result.

       We added the statement “These selected values of the virtual spring constant, pull velocity and tensile forces were moderate and could cause mechano-destabilization of the complex either in the early loading phase at “force-ramp” mode or in entire simulation duration at “force-clamp” mode” into the Material and Method section in the revised manuscript. (the last paragraph in the page 10 in the revised manuscript)

Comment 2. Are there experimental validations or simulational validations using a simpler but well-studied system?

Answer:  In spite of lack of experimental and simulative validations on the present molecular system, some experimental and simulative validations from different systems, such as P-, E-, or L-selectin with PSGL-1 [21,26,27], von Willebrand factor with GPIbα [20], ADMAMTS13 (A Disintegrin and Metalloprotease with thrombos-pondin motifs-13)[28], and CD44-HA interaction (Yao, Z.; Wu, J.; Fang, Y. Moderate Constraint Facilitates Association and Force-Dependent Dissociation of HA-CD44 Complex. IJMS 2023, 24) as well as Mac-1/GPIba interaction (Jiang, X.; Sun, X.; et al. MD Simulations on a Well-Built Docking Model Reveal Fine Mechanical Stability and Force-Dependent Dissociation of Mac-1/GPIba Complex. Front Mol Biosci 2021, 8, 638396), should support our present results.

Comment 3. On page 10, the subsubsection numbers do not match the subsection number. For example, 3.1, 3.2 are under subsection 4. Materials and Methods

Answer: We are sorry for these mistakes. We have corrected them in the revised manuscript.

Comment 4. There are some grammar errors in the manuscripts. For example, this sentence: “However, less knowledge on (of) mechanical regulation on DNAM-1 interaction with CD155 exists, despite the DNAM-1-related events usually occurs (occur) at diverse mechanical micro-environments”. And there are a few places where present and past tense are used inconsistently. I would suggest the author proofread the manuscript more carefully and correct those errors.

Answer: We are sorry for these mistakes. We have corrected them in the revised manuscript.

Round 2

Reviewer 1 Report

I can accept the answer for my and other reviewer's answer. The paper can be accepted.